# Mo-Doped Na_4_Fe_3_(PO_4_)_2_P_2_O_7_/C Composites for High-Rate and Long-Life Sodium-Ion Batteries

**DOI:** 10.3390/ma17112679

**Published:** 2024-06-01

**Authors:** Tongtong Chen, Xianying Han, Mengling Jie, Zhiwu Guo, Jiangang Li, Xiangming He

**Affiliations:** 1College of New Materials and Chemical Engineering, Beijing Institute of Petrochemical Technology, Beijing 102617, China; ctt10105204@163.com (T.C.); hanxianying@bipt.edu.cn (X.H.); jml16953708@163.com (M.J.); guozhiwu@bipt.edu.cn (Z.G.); 2Beijing Key Laboratory of Fuels Cleaning and Advanced Catalytic Emission Reduction Technology, Beijing 102617, China; 3Institute of Nuclear & New Energy Technology, Tsinghua University, Beijing 100084, China

**Keywords:** sodium-ion batteries, cathode materials, polyanionic, Mo^6+^ doping, DFT calculation

## Abstract

Na_4_Fe_3_(PO_4_)_2_P_2_O_7_/C (NFPP) is a promising cathode material for sodium-ion batteries, but its electrochemical performance is heavily impeded by its low electronic conductivity. To address this, pure-phase Mo^6+^-doped Na_4_Fe_3−x_Mo_x_(PO_4_)_2_P_2_O_7_/C (Mox-NFPP, x = 0, 0.05, 0.10, 0.15) with the Pn21a space group is successfully synthesized through spray drying and annealing methods. Density functional theory (DFT) calculations reveal that Mo^6+^ doping facilitates the transition of electrons from the valence to the conduction band, thus enhancing the intrinsic electron conductivity of Mox-NFPP. With an optimal Mo^6+^ doping level of x = 0.10, Mo0.10-NFPP exhibits lower charge transfer resistance, higher sodium-ion diffusion coefficients, and superior rate performance. As a result, the Mo0.10-NFPP cathode offers an initial discharge capacity of up to 123.9 mAh g^−1^ at 0.1 C, nearly reaching its theoretical capacity. Even at a high rate of 10 C, it delivers a high discharge capacity of 86.09 mAh g^−1^, maintaining 96.18% of its capacity after 500 cycles. This research presents a new and straightforward strategy to enhance the electrochemical performance of NFPP cathode materials for sodium-ion batteries.

## 1. Introduction

The development of new energy and energy storage technologies is crucial in implementing the green development and carbon reduction strategy [1,2]. However, the scarcity of lithium resources makes it challenging for lithium-ion batteries to meet the growing demand for energy storage [3]. In contrast, sodium resources are abundant and inexpensive, positioning sodium-ion batteries as a promising alternative for widespread use in energy storage [4,5]. These batteries utilize cathode materials such as layered transition metal oxides, Prussian blue analogs, and polyanion compounds [6,7,8]. Among these, the polyanionic sodium-ion battery cathode material Na_4_Fe_3_(PO_4_)_2_P_2_O_7_/C (NFPP) stands out due to its high theoretical specific capacity of 129 mAh g^−1^ and stable discharge voltage of 3.2 V. Additionally, it exhibits minimal volume changes during the charging and discharging process, ensuring good cycling stability and safety. Despite these advantages, its insufficient conductivity hampers its electrochemical performance and rate capability, thereby limiting its widespread application [8,9].

To enhance the electronic conductivity and ionic diffusion in NFPP, researchers have explored several methods, including carbon coating [10], ion doping [11,12,13,14,15,16,17,18,19], and nanostructure and morphology control [20]. In particular, ion doping, such as with Mn^2+^ [11,12], Mg^2+^ [13], Cr^3+^ [14], Ti^4+^ [15], V^3+^ [16], Mn-F dual-elements [17], and high-entropy multi-elements [18,19], has proven to be highly effective. Mn^2+^ doping, as shown by Li et al., enhances the Na^+^ diffusion kinetics, conductivity, and structural stability of NFPP/rGO composites [11]. Similarly, Tao et al. confirmed the benefits of Mn doping [12]. Xiong et al. demonstrated that 5% Mg^2+^ doping improved the Na^+^ diffusion and reduced the charge transfer resistance, enhancing the rate capability and cycle life [13]. Huang et al. found that Ti^4+^ doping created vacancies that facilitated ion and electron transport, significantly improving the low-temperature performance [15]. Zhang et al. showed that V^3+^ doping lowered the Na^+^ migration energy barrier, enhancing the ion and electronic conductivity and structural stability. Recent studies on Mn-F dual-element doping and high-entropy multi-element doping [17] with Na_4_Fe_2.95_(NiCoMnMgZn)_0.01_(PO_4_)_2_P_2_O_7_ [18] and Na_4_Fe_2.85_(NiCoMnCuMg)_0.03_(PO_4_)_2_P_2_O_7_ [19] have shown potential to improve the high-rate and long-cycle performance. 

Doping with high-valence Mo^6+^ has been proven to significantly enhance the performance of various lithium-/sodium-ion battery cathode materials, including Li_3_V_2_(PO_4_)_3_/C [21], LiNi_0.6_Co_0.2_Mn_0.2_O_2_ [22], LiNi_0.5_Mn_1.5_O_4_ [23], Na_3_V_2_(PO_4_)_3_@C [24,25,26], Li_1.2_Ni_0.13_Fe_0.13_Mn_0.54_O_2_ [27], and LiMn_0.6_Fe_0.4_PO_4_ [28], etc. For example, Wen et al. [28] enhanced the Li^+^ diffusion rate in LiMn_0.6_Fe_0.4_PO_4_ materials by doping with Mo^6+^, achieving discharge capacities of 153.2 mAh g^−1^ at 0.2 C and 94.2 mAh g^−1^ at 10 C, with a capacity retention rate of 91.4% after 100 cycles at 1 C. For polyanionic cathode materials, the performance enhancement achieved by Mo^6+^ doping is not only attributed to the strong Mo-O bond energy (502 kJ/mol) [29], which can increase the structural stability, but also to the decrease in charge transfer impedance and the generation of vacancies, which enhance the ion/electron transport, thereby accelerating the ion transport rates [21,24,26]. Additionally, Kumar et al. demonstrated that Mo doping in trimetallic oxides significantly enhances the redox activity by optimizing the energy barriers in the reaction steps and increasing the adsorption energy of intermediates, thereby improving the overall performance in zinc–air batteries [30]. However, studies on Mo^6+^ doping in NFPP have not yet been reported. 

Herein, we present a straightforward method for the synthesis of Mo^6+^-doped Na_4_Fe_3−X_Mo_x_(PO_4_)_2_P_2_O_7_/C composites (Mox-NFPP; x = 0, 0.05, 0.10, 0.15) as cathode materials for sodium-ion batteries. We investigated the effects of Mo^6+^ doping on the material’s structure and performance, finding that the optimized composition significantly enhanced the charge–discharge performance. The Mo0.10-NFPP composite exhibited excellent rate capability, achieving 86.09 mAh g^−1^ even at a high 10 C rate, with capacity retention of 96.18% after 500 cycles. 

## 2. Materials and Methods

### 2.1. Synthesis of Mox-NFPP Materials

Mox-NFPP nanocomposites were synthesized via a spray drying process, using raw materials including CH_3_COONa·3H_2_O (Beijing Chemical Works, Beijing, China, 99.7%), Fe(NO_3_)_3_·9H_2_O (Sinopharm Chemical Reagent Co., Ltd., Shanghai, China, 99.7%), NH_4_H_2_PO_4_ (Xilong Scientific Co., Ltd., Shantou, China, 99.7%), citric acid (C_6_H_8_O_7_·H_2_O) (Xilong Scientific Co., Ltd., Shantou, China, 99.7%), Mo(NH_4_)_6_·4H_2_O (Beijing Institute of Chemical Reagents Co., Ltd., Beijing, China, 99.7%), and oxalate (C_2_H_2_O_4_·2H_2_O) (Beijing Institute of Chemical Reagents Co., Ltd., Beijing, China, 99.7%). The materials, with varying Mo content (x = 0, 0.05, 0.10, and 0.15), were designated as 0-NFPP, Mo0.05-NFPP, Mo0.10-NFPP, and Mo0.15-NFPP, respectively. For instance, the preparation process for 0-NFPP involved dissolving a mixture of 20 mmol CH_3_COONa·3H_2_O, 15 mmol Fe(NO_3_)_3_·9H_2_O, 20 mmol NH_4_H_2_PO_4_, 11.25 mmol C_6_H_8_O_7_·H_2_O, and 45 mmol C_2_H_2_O_4_·2H_2_O in 50 mL deionized water through magnetic stirring. The solution was then spray-dried using an HZ-1500 experimental spray dryer (Shanghai Huizhan Experimental Equipment Co., Ltd., Shanghai, China), with an inlet temperature 220 °C, an outlet temperature ranging from 90 to 109 °C, a peristaltic pump speed of 23 r/min, a fan frequency of 40 Hz, and a needle interval of 2 s. Following spray drying, the precursor was calcined in an OTF1200X-II tube furnace (Shanghai Meicheng Scientific Instruments Co., Ltd., Shanghai, China) filled with argon at a rate of 5 °C/min, initially at 300 °C for 3 h, and subsequently heated at 550 °C for 10 h. After cooling, the precursor was ground and sieved to obtain the final product.

### 2.2. Microstructural Characterization

The crystal phase and valence state analysis were performed using an Aeris-type X-ray diffractometer (Malvern Instruments, Malvern, UK) and a Nexsa G2-type X-ray photoelectron spectrometer (Thermo Scientific, Waltham, MA, USA), respectively. The structure and morphology of the materials were then examined with a JEM-F200 field emission transmission electron microscope (JEOL, Tokyo, Japan). Subsequently, surface morphology and element mapping characterization were carried out using a Quanta 400 field emission electron microscope (FEI, Omaha, NE, USA). Finally, the carbon content in the samples was determined using an SDT650 thermogravimetric/differential thermal analyzer (TA Instruments, New Castle, DE, USA).

### 2.3. Electrochemical Measurements

The electrode slurry was prepared using an active material, carbon black (China Grinm Group Co., Ltd., Beijing, China), PVDF (Guangdong Canrd New Energy Technology Co., Ltd., Dongguan, China, 99.7%), and N-methyl-2-pyrrolidone solvent (Aladdin Holdings Group Co., Ltd., Shanghai, China, 99.7%), in which the mass ratio of active material–carbon black–PVDF was controlled to be 8:1:1. The active substance weighed 0.4 g. This slurry was coated onto aluminum foil, targeting an active material loading of 6–8 mg cm^−2^. After drying, the coated foil was pressed and cut into 8 mm diameter discs, weighed, and assembled with a sodium disc (Shenzhen Kejing Star Technology Company, Shenzhen, China), glass fiber separator (Whatman, Buckinghamshire, UK), and 1 mol L^−1^ NaPF_6_/EC + DEC (volume ratio 1:1) electrolyte (Guangdong Canrd New Energy Technology Co., Ltd., Dongguan, China) into a CR2032 coin test cell in an argon-filled glove box. Electrochemical tests were conducted using a CT-2001A LAND battery tester (Wuhan Jinnuo Electronics Co., Ltd., Wuhan, China), assessing the rate capability (1 C = 129 mAh g^−1^) and cycling performance within a voltage range of 1.7–4.3 V. The sodium-ion diffusion coefficients were determined through the galvanostatic intermittent titration technique (GITT) [11,31] by cycling the battery at a 0.1 C rate over the voltage range of 1.7–4.3 V, with pulse durations of 30 min and relaxation times of 10 min. After three cycles at 0.1 C and a subsequent 2 h rest period, electrochemical impedance spectroscopy (EIS) measurements were performed using an IM6eX electrochemical workstation (ZAHNER, Kronach, Germany) over a frequency range of 100 mHz to 100 kHz and at a voltage amplitude of 5 mV. The same setup was used to evaluate the temperature-dependent impedance at temperatures of 30 °C, 35 °C, 45 °C, 55 °C, and 65 °C.

### 2.4. First-Principles Calculation Method

The density of states calculation was conducted using the projected augmented wave method within the framework of density functional theory, employing the VASP 5.4.1 software. The computational model utilized a supercell of dimensions (a × b × c) containing four basic units of Na_4_Fe_3−x_Mo_x_(PO_4_)_2_(P_2_O_7_) (x = 0, 0.08). To address the strong electron correlation in d-electrons, the GGA + U method was applied, with the U_eff_ values for Fe and Mo d-electrons set at 4.3 and 6.3, respectively, as referenced in the literature [32,33,34,35,36,37]. During the structure optimization, the convergence criteria for energy and forces were established at 1.0 × 10^−5^ eV and 0.02 eV/Å, respectively. Integration over the Brillouin zone was executed using a 2 × 5 × 3 Monkhorst-Pack grid.

## 3. Results and Discussion

### 3.1. Morphology and Structural Characterization 

Mox-NFPP nanocomposites were synthesized using a spray drying process, as illustrated in Figure 1. The spray drying technique can ensure the homogeneous mixture of the raw materials, which provides a guarantee for the preparation of pure-phase sodium-ion battery materials with high electrochemical performance.

Figure 2a,b display the transmission electron microscopy (TEM) and high-resolution transmission electron microscopy (HRTEM) images of the prepared NFPP material. As illustrated in Figure 2a, the NFPP material comprises nanoparticles smaller than 50 nm. This nanostructure may be related to the inhibition of grain growth by the appropriate carbon coating (carbon content of 3.8 wt%, Appendix A), which can reduce the diffusion distance of Na^+^ within the material, thus enhancing its charge–discharge performance. Furthermore, the lattice fringes corresponding to the (200) crystal planes are clearly visible in the HRTEM image (Figure 2b), indicating the high crystallinity of the NFPP material. Scanning electron microscopy (SEM) was employed to analyze the morphological features of the 0-NFPP, Mo0.05-NFPP, Mo0.10-NFPP, and Mo0.15-NFPP materials with varying Mo^6+^ doping levels, as shown in Figure 2c–f. The analyses reveal that all materials possess a porous tunnel structure, and the porosity increases as the Mo^6+^ doping level rises, thereby enhancing the contact area with the electrolyte and being beneficial in improving the material’s performance. Additionally, the energy-dispersive spectroscopy (EDS) analysis of the Mo0.10-NFPP sample (Figure 2g) demonstrates the uniform distribution of all elements in the Mo-doped materials.

X-ray diffraction (XRD) was utilized to investigate the effect of Mo^6+^ doping on the crystal structure of the material, as illustrated in Figure 3a. The XRD diffraction peak positions for both the undoped (0-NFPP) and doped materials are consistent with the NFPP standard card (PDF#89-0579), all belonging to the orthorhombic structure of the Pn21a space group [15]. The absence of other diffraction peaks confirms that the materials are single-phase. Compared to 0-NFPP, all of the diffraction peaks of Mox-NFPP shift towards lower angles. For instance, the (022) crystal plane’s characteristic peak in 0-NFPP is at 32.2°, while, in Mo1.5-NFPP, it shifts to 31.9° (Figure 3b). Bragg’s Law suggests that this shift towards smaller angles indicates the expansion of the lattice spacing. Given the smaller radius of Mo^6+^ ions (0.59 Å) compared to Fe^2+^ ions (0.78 Å), it is likely that the increased lattice spacing is due to non-equivalent substitution by Mo^6+^ doping. This results in the creation of vacancies for charge balance [26] and an increase in the interplanar crystal spacing, which could improve the Na^+^ ion transport. In addition, the XRD of Mox-NFPP was subjected to Rietveld refinement, with the fitting results displayed in Appendix A. The lattice parameters of *b* and volume (*V*) of the Pn21a phases for Mox-NFPP both increased compared to the pristine 0-NFPP. These results further support the process by which Mo^6+^ ions enter the crystal lattice, leading to its expansion.

To ascertain the valence states of the elements in the Mo0.10-NFPP material, X-ray photoelectron spectroscopy (XPS) analysis was performed. As shown in Appendix A, the XPS survey confirms the presence of Na, Fe, O, P, and Mo elements. This aligns well with the EDS mapping results showcased in the above Figure 2g. Appendix A show that the Na 1s, O 1s, and P 2p spectra are consistent with the XPS fitting results of the NFPP material as detailed in the literature [38]. As reported by Zhao [39], the Fe^2+^ orbital splitting peaks are located at 725.4 and 711.4 eV in the pristine NFPP material. Here, in the Mo0.10-NFPP material, the Fe 2p peaks shift marginally to the lower binding energy levels of 724.71 eV and 711.23 eV (Figure 3c). This suggests that the divalent state of iron in Mo0.10-NFPP remains constant after Mo^6+^ doping. The Fe-O bonds are slightly weakened, due to the fact that Mo atoms have a stronger ability to confer a charge to nearby O atoms than Fe atoms. In Figure 3d, the main peaks corresponding to Mo 3d_5/2_ and Mo 3d_3/2_ are observed at 232.41 eV and 235.05 eV, respectively, confirming that the Mo element is in the +6 oxidation state [24,26], which means that the Mo^6+^ ions do not participate in the electrochemical reactions during the charging process. This result is consistent with the absence of a new voltage plateau in the first charge–discharge curve of the Mo^6+^-doped material shown in Figure 4a.

### 3.2. Electrochemical Performance

A series of electrochemical performance tests were carried out on the Mox-NFPP materials, as illustrated in Figure 4. Figure 4a presents the initial charge–discharge curves at a 0.1 C rate, where the four samples demonstrate similar profiles, featuring three pairs of redox plateaus at approximately 2.7 V, 3.0 V, and 3.2 V. These plateaus correspond to the sequential release of Na^+^ ions from the Na3, Na1, and Na4 sites, as referenced in previous studies [12,16,18,19]. The discharge capacities of the 0-NFPP, Mo0.05-NFPP, Mo0.10-NFPP, and Mo0.15-NFPP samples were 105.8, 120.5, 123.9, and 117.2 mAh g^−1^, respectively. The samples doped with Mo^6+^ displayed reduced charge–discharge polarization and achieved higher capacities compared to the undoped sample. Furthermore, the Mo0.10-NFPP sample demonstrated superior rate performance, achieving a specific discharge capacity of 95.71 mAh g^−1^ at a 5 C rate, as detailed in Figure 4b.

Electrochemical impedance spectroscopy (EIS) tests were performed on the synthesized Mox-NFPP materials, as illustrated in Figure 4c. The equivalent circuit model is included in Figure 4c, where *R_s_* represents the ohmic resistance, and *R_ct_* reflects the charge transfer resistance. The semicircle in the high-frequency region of the EIS curves corresponds to the *R_ct_* element, whereas the slope in the low-frequency region is dictated by the Warburg diffusion of Na^+^ ions from the surface to the middle of the cathode particles. The fitting results outlined in Appendix A highlight that *R_s_* decreases slightly when increasing the Mo^6+^ doping level in the Mox-NFPP samples. However, *R_ct_* shows a significant drop. This decrease in *R_ct_* is linked to the non-equivalent substitution of Mo^6+^, causing vacancies that speed up the transport of ions and electrons, thus enhancing the charge transfer reactions [21,24,26]. The decrease in *R_ct_* contributes to the improved capacity and rate performance of the Mox-NFPP samples. However, it is worth noting that the *R_ct_* for Mo0.15-NFPP is higher than that for Mo0.10-NFPP. This could be due to the inactivity of Mo^6+^ ions during the charge–discharge processes, potentially causing the performance to drop with excessive doping. Further cycling performance analysis (Figure 4d) demonstrates that Mo0.10-NFPP achieves a capacity of 86.09 mAh g^−1^ at a 10 C rate. Even after 500 cycles, it retains 96.18% of its capacity, demonstrating its excellent structural stability. Compared to other ion-doped NFPP materials reported recently, Mo0.10-NFPP exhibits better electrochemical performance than Mn^2+^ [12], Mg^2+^ [13], Cr^3+^ [14], and Ti^4+^ [15]-doped materials (Appendix A), in which the composite carbon materials are produced from glucose, sucrose, or citric acid pyrolysis. V^3+^ doping also shows a better effect in improving the rate performance of the NFPP material [16], but its high toxicity limits its application. Although composite rGO or CNT can enable Mn^2+^ [11] and Cr^3+^ [14]-doped NFPP materials to achieve remarkably improved rate performance, the expensive rGO and CNT will significantly increase the battery costs. Therefore, for NFPP materials, Mo^6+^ doping should be an effective modification method, with promising application prospects and feasibility.

The sodium-ion diffusion coefficients of the Mox-NFPP materials were determined using the galvanostatic intermittent titration technique (GITT) [11,31], as depicted in Figure 5a and Appendix A. In the discharge voltage range of 2–3.5 V, the average diffusion coefficients (*D*) were 1.92 × 10^−10^, 2.04 × 10^−10^, 2.45 × 10^−10^, and 2.03 × 10^−10^ cm^2^ s^−1^ for the 0-NFPP, Mo0.05-NFPP, Mo0.10-NFPP, and Mo0.15-NFPP material, respectively. Notably, the Na^+^ diffusion coefficient increased as the Mo^6+^ doping level x was raised to 0.1. Thus, the Mo0.10-NFPP sample exhibited the highest diffusion coefficient, aligning with its superior electrochemical performance. The activation energy, an important parameter reflecting Na^+^ transport [39,40], was also measured for the 0-NFPP and Mo0.10-NFPP samples. Using the measured charge transfer resistance (*R*_ct_) at various temperatures, as shown in Figure 5b,c, the ionic conductivity (*σ*) was calculated using the formula *σ* = *l*/(*A* × *R*_ct_) (where *σ* is the ionic conductivity, *l* is the electrode thickness, and *A* is the electrode area), and the corresponding ionic conductivity *σ* was calculated. The activation energy (*E_a_*) of the materials was then obtained from the Arrhenius plot (ln(*σT*)~1000/*T*) shown in Figure 5d [41]. The activation energy of the 0-NFPP sample was measured to be 40.94 kJ mol^−1^, while the activation energy of the Mo0.10-NFPP material decreased to 34.77 kJ mol^−1^, indicating that Mo doping reduced the ion transport energy barrier, which is beneficial in accelerating Na^+^ ion diffusion.

To further investigate the impact of Mo^6+^ ion doping on the conductivity of the material, the structure of the material before and after Mo^6+^ doping was studied using first-principles calculations. Following structural optimization, a doping model was constructed by substituting a Mo atom for any Fe atom at the Fe1 or Fe2 positions (Figure 6a). The calculated density of states shows that NFPP (Figure 6b) has a large bandgap (about 3.1 eV) and a conduction band with an energy level difference of about 2.9 eV from the Fermi level, which is consistent with the literature [40]. In contrast, the conduction band of Mo^6+^-doped NFPP (Figure 6c), which primarily consists of the non-bonding orbitals from Fe’s d orbitals, displays a reduced energy gap of 1.0 eV to the Fermi level, indicating typical n-type doping by Mo. This conclusion is further corroborated by the Bader charge analysis. In the Mo-doped NFPP model, the average charge of the six O atoms near the Mo atom is −1.4856 e, while other O atoms not directly bonded to Mo have an average charge of only −1.4789 e, indicating that Mo atoms have a stronger ability to confer a charge to nearby O atoms than Fe atoms. Consequently, the impurity level formed by the interaction of the Mo atom’s 4d orbitals with the surrounding O atom’s 2p orbitals is at a higher energy level close to the Fermi level, approximately 1.1 eV higher than the level formed by the interaction of the Fe atom’s 3d orbitals with the surrounding O atom’s 2p orbitals. Although Mo doping slightly reduces the bandgap width, the n-type doping by Mo atoms significantly shifts the Fermi level towards the conduction band, and it also forms the highest occupied impurity states near the Fermi level, which facilitates the transition of electrons from the valence band to the conduction band, thereby enhancing the conductivity. This is consistent with the previously mentioned significant improvement in the electrochemical performance of Mo^6+^-doped NFPP materials.

## 4. Conclusions

Pure-phase Mo^6+^-doped Mox-NFPP (x = 0, 0.05, 0.10, 0.15) composites were successfully synthesized using the spray drying method. The appropriate doping of Mo^6+^ ions not only promoted the transition of electrons from the valence band to the conduction band, enhancing the intrinsic electron conductivity, but also reduced the charge transfer resistance and accelerated the Na^+^ ion transport, thereby improving the electrochemical performance of the materials. Notably, the sample with a Mo^6+^ doping level of x = 0.10 demonstrated the most impressive electrochemical properties, achieving a discharge capacity of 123.9 mAh g^−1^ at a 0.1 C rate and maintaining 86.09 mAh g^−1^ even at a high 10 C rate, with capacity retention of 96.18% after 500 cycles. These findings affirm that Mo^6+^ ion doping is a potent strategy to enhance the high-rate charge–discharge capabilities of NFPP, offering a practical method for the development of high-performance polyanionic cathode materials for sodium-ion batteries.

## Figures and Tables

**Figure 1 materials-17-02679-f001:**
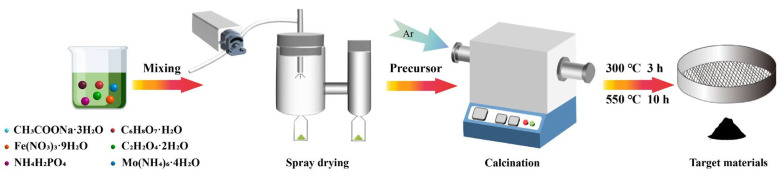
Schematic of the spray drying synthesis of Mox-NFPP nanocomposites.

**Figure 2 materials-17-02679-f002:**
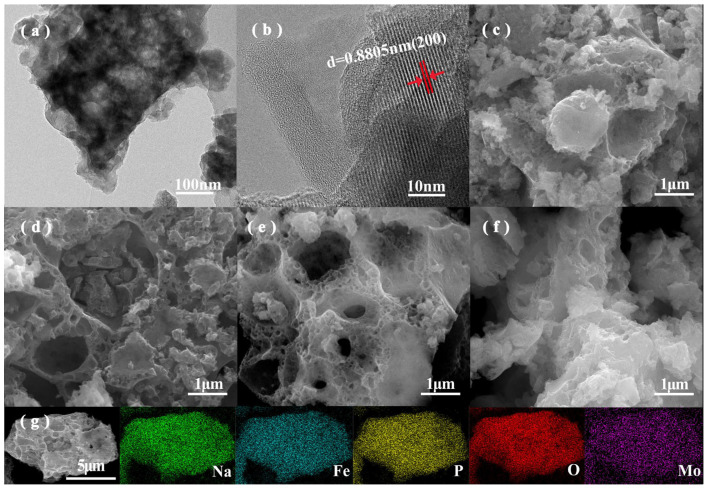
(**a**,**b**) TEM and HRTEM images of 0-NFPP. (**c**–**f**) SEM images of 0-NFPP, Mo0.05-NFPP, Mo0.10-NFPP, Mo0.15-NFPP. (**g**) EDS mapping images of Mo0.10-NFPP material.

**Figure 3 materials-17-02679-f003:**
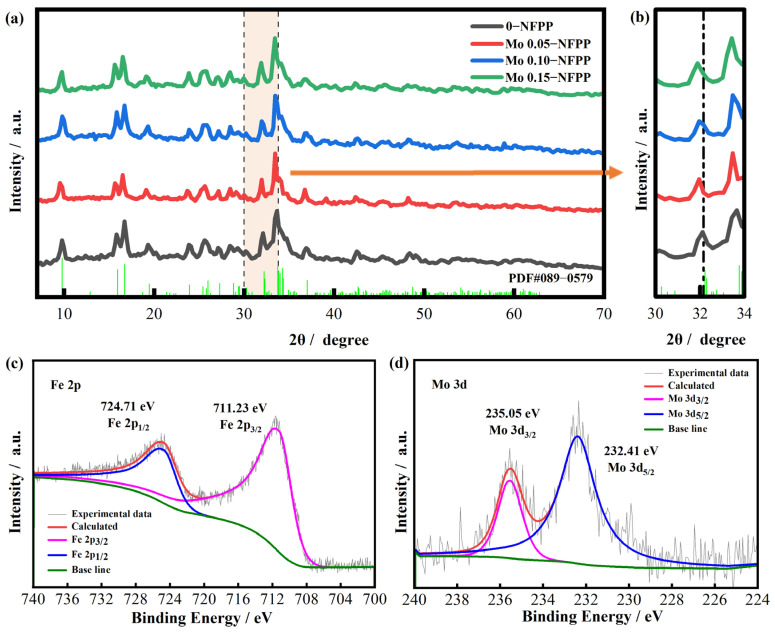
(**a**,**b**) XRD patterns of Mox-NFPP materials. (**c**) Fe 2p XPS spectrum and (**d**) Mo 3d XPS spectrum of Mo0.10-NFPP material.

**Figure 4 materials-17-02679-f004:**
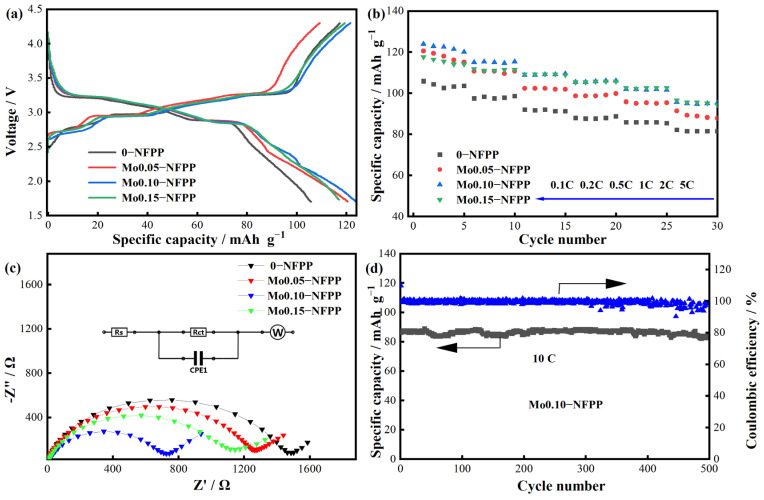
(**a**) First charge–discharge curves at 0.1 C for Mox-NFPP materials. (**b**) Rate performance of Mox-NFPP materials. (**c**) Nyquist plots of EIS and equivalent circuit for Mox-NFPP materials. (**d**) Cycling performance of Mo0.10-NFPP material at 10 C rate.

**Figure 5 materials-17-02679-f005:**
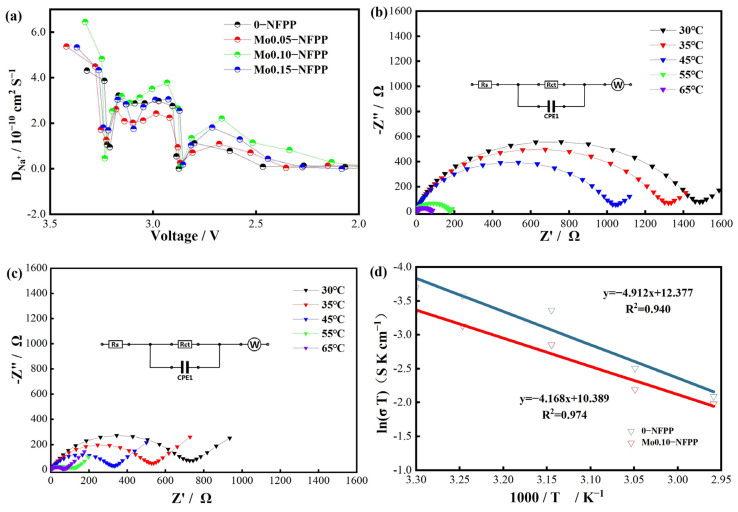
(**a**) Na^+^ diffusion coefficients of Mo-doped NFPP. (**b**,**c**) Variable-temperature impedance plots of 0-NFPP and Mo0.10-NFPP and (**d**) corresponding Arrhenius plots.

**Figure 6 materials-17-02679-f006:**
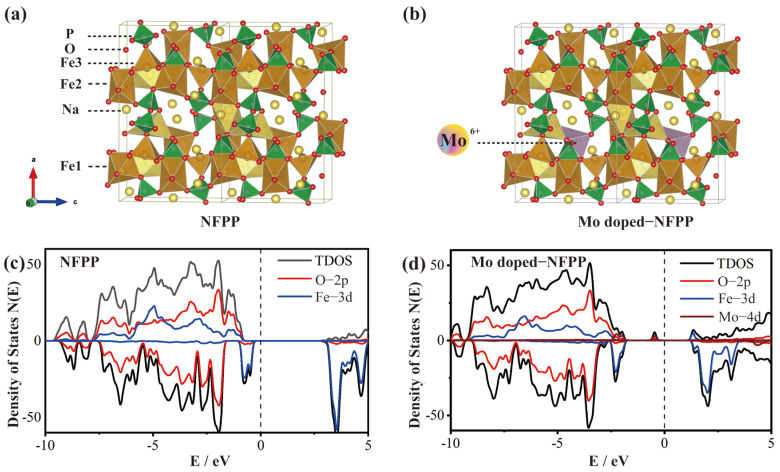
(**a**,**b**) Crystal structure of 0-NFPP and Mo-doped NFPP. (**c**,**d**) Density of states of 0-NFPP and Mo-doped NFPP.

## Data Availability

The original contributions presented in the study are included in the article/Appendix A, further inquiries can be directed to the corresponding authors.

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
