# Peer review of "Mo-Doped Na4Fe3(PO4)2P2O7/C Composites for High-Rate and Long-Life Sodium-Ion Batteries"

_materials, 2024, doi:10.3390/ma17112679_

Round 1

Reviewer 1 Report

Comments and Suggestions for Authors

Dear authors,

Thank you for presenting this well-prepared manuscript to Materials. I recommend to publish the manuscript with only one very small correction: Please give the space group symbol in the proper form. This means all letters in italics and the numbers with subscript: Pna21. Why has the non-standard setting Pn21a been chosen?

Sincere greetings

Author Response

Thank you very much for your comments and suggestion. As suggested, we have standardized the space group of Pna21 with highlight marks in pages 1 and 5.

Reviewer 2 Report

Comments and Suggestions for Authors

Authors reported the effect of Mo doping in the performance of Na-ion batteries based on Na4Fe3(PO4)2P2O7/C composites. Adequate Mo6+ doping exhibited lower charge transfer resistance, higher sodium ion diffusion coefficients and rate performance. This manuscript can be accepted for publication after following revisions:

1.      The abstract section should be focused to the major finding and objectives. The introduction section is prolix and still the literature review for the effect of Mo dopants can be explained with following reference:  doi.org/10.1002/advs.202303525

2.      Provide the exact amount of electrode materials used to make electrode slurry. Indicate the percentage purity of the chemical used.

3.      Update the revised manuscript with the HR XPS of Fe 2p, O1s and P 2p also.

4.      Is there any change in the size of crystal upon Mo6+ doping?

Comments on the Quality of English Language

 Minor editing of English language required

Author Response

Thank you very much for taking the time to review our manuscript. Please find the detailed responses below and the corresponding revisions in the re-submitted files. 

1. The abstract section should be focused to the major finding and objectives. The introduction section is prolix and still the literature review for the effect of Mo dopants can be explained with following reference: doi.org/10.1002/advs.202303525

Response 1: Thank you for your suggestion. In the main manuscript, we have revised the abstract to emphasize the highlights of this research as following, and refined the introduction section for clarity and efficiency, including the reference [30] of Adv. Sci. 2023, 10, 2303525. All the revisions have been marked in yellow color in the main manuscript.

"Abstract: Na4Fe3(PO4)2P2O7/C (NFPP) is a promising cathode material for sodium-ion batteries, but its electrochemical performance is heavily impeded by its low electronic conductivity. To address this, pure-phase Mo6+-doped Na4Fe3-xMox(PO4)2P2O7/C (Mox-NFPP, x = 0, 0.05, 0.10, 0.15) with Pna21 space group are successfully synthesized through spray drying and annealing methods. Density functional theory (DFT) calculations reveal that Mo6+ doping facilitates the transition of electrons from the valence to the conduction band, thus enhancing the intrinsic electron conductivity of Mox-NFPP. With an optimal Mo6+ doping level of x = 0.10, the Mo0.10-NFPP exhibits lower charge transfer resistance, higher sodium ion diffusion coefficients, and superior rate performance. As a result, the Mo0.10-NFPP cathode offers an initial discharge capacity up to 123.9 mAh g−1 at 0.1 C, nearly reaching its theoretical capacity. Even at a high rate of 10 C, it delivers a high discharge capacity of 86.09 mAh g−1, maintaining 96.18% of its capacity after 500 cycles. This research presents a new and straightforward strategy for enhancing the electrochemical performance of NFPP cathode materials for sodium-ion batteries."

2. Provide the exact amount of electrode materials used to make electrode slurry. Indicate the percentage purity of the chemical used.

Response 2: Thank you for your valuable advice. As suggested, we've included the quantities of active materials, conductive carbon, and binder used for electrode fabrication on page 3, lines 118-123, and the percentage purity of the chemical agents are also given in the experiment section (page 3).

3. Update the revised manuscript with the HR XPS of Fe 2p, O1s and P 2p also.

Response 3: Thank you for your suggestion. As suggested, the HR XPS data of Fe 2p, O 1s and P 2p in Mo0.10-NFPP sample is analyzed and added in pages 5 and 6. Specifically, in Figure 3c of Fe 2p spectra, the characteristic peak at 711 eV belongs to Fe2+, from the reduction of Fe(NO3)3·9H2O during the synthesis process (Nano Energy, 2022, 91, 106680-106689). In Figure S3c, O1s spectra show one strong and two weak peaks at 531.09, 532.76, and 535.74 eV, which are attributed to O-P, O-C, and O=C-O bonds, respectively. In Figure S3d, P 2p spectra with characteristic peak at 133.5 eV is also consistent with the results of NFPP in related literature (ChemSusChem. 2021, 14, 5424).

4. Is there any change in the size of crystal upon Mo6+ doping?

Response 4: Thank you for your question. In the revised manuscript, we have added the lattice parameter data obtained from XRD refinement and the related discussions on pages 5, lines 182-193. According to the XRD results in Figure 3a, all the diffraction peaks of Mox-NFPP shift towards lower angels than 0-NFPP. In the partial magnification of Figure 3b, the (022) crystal plane’s characteristic peak in 0-NFPP is at = 32.2°, while in Mo1.5-NFPP it shifts to = 31.9° (Figure 3b). According to Bragg's Law, this shift towards smaller angles indicates an expansion of the lattice spacing. Given the smaller radius of Mo6+ ions (0.59 Å) compared to Fe2+ ions (0.78 Å), it is likely that the increase lattice spacing is due to non-equivalent substitution by Mo6+ doping. This results in the creation of vacancies for charge balance, and the increase of interplanar crystal spacing, which could improve Na+ ion transport. In addition, the XRD of Mox-NFPP was subjected to Rietveld refinement, with the fitting results displayed in Figure S2a-S2d and Table S1. The lattice parameters of b and volume (V) of the Pna21 phases for Mox-NFPP have both increased compared to the pristine 0-NFPP. These results further support the process that Mo6+ ions enter the crystal lattice, leading to its expansion.

Reviewer 3 Report

Comments and Suggestions for Authors

materials-3012023

Comments

The presented article reports the rational synthesis of Na4Fe3-xMox(PO4)2P2O7/C (x = 0, 0.05, 0.10, 0.15) using spray drying and followed by annealing methods for long-life sodium-ion batteries. The as-obtained materials were analyzed using various techniques such as SEM, TEM, XRD, and XPS, along with the needed electrochemical tests. The article is well organized and could be accepted after addressing the following comments:

  1. The authors should rewrite the abstract and focus to emphasize the main findings and novelty of this study. Also, the abbreviations should be defined.
  2. The TOC does not reflect the novelty and main findings of this study please modfy.
  3. The resolution of all figures should be improved significantly
  4. The authors should rewrite the introductions section and improve the literature review. Also there are many sentences without cited references. The objectives of this study should be emphasized at the end of the introduction

5.     The authors should carefully revise the scale bars, magnifications and spelling in Figure 2. For example ‘’um’’ should be ‘’µm’’. Also, please revise the materials abbreviations in the caption

6.     The authors should mention the source of materials and precursors  

7.     In Figure 3a, at the x-axis, please replace ‘’o’’ with ‘’angle’’. also the discussion related to the XRD data should be improved significantly and effect of doping should be highlighted. The diffraction peaks are the same with only slight shift, please explain  

  1. The authors should add the high-resolution XPS spectra of all elements and fit them to see the effect of doping on the valence states and show the element phases
  2. A comparison table for the performance with previously published articles should be added.
  3. The explanation and discussions related to EIS data should be revised carefully. For instance ‘’ Figure 4c, the charge transfer resistances (Rct) for’’ It is Nyquist plot. Also the authors should fit the EIS data and add the solution resistance (Rs), charge transfer resistance (Rct), Warburg impedance (Wd), and constant phase element (CPE).
  4. The literature review is not enough, and more references should be cited, like ‘’ Nanomaterials 2022, 12(16), 2825 & Energy Conversion and Management: X, 2024. 22: p. 100570

Comments on the Quality of English Language

Moderate editing of English language required

Author Response

Thank you very much for taking the time to review our manuscript. Please find the detailed responses below and the corresponding revisions in the re-submitted files. The modified parts have been marked yellow color in the revised manuscript. 

1. The authors should rewrite the abstract and focus to emphasize the main findings and novelty of this study. Also, the abbreviations should be defined.

Response 1: Thank you for your suggestion. In the main manuscript, we have revised the abstract to emphasize the highlights of this research as following. Additionally, we've made corrections and standardized the abbreviations for NFPP and Mox-NFPP materials.

“Abstract: Na4Fe3(PO4)2P2O7/C (NFPP) is a promising cathode material for sodium-ion batteries, but its electrochemical performance is heavily impeded by its low electronic conductivity. To address this, pure-phase Mo6+-doped Na4Fe3-xMox(PO4)2P2O7/C (Mox-NFPP, x = 0, 0.05, 0.10, 0.15) with Pna21 space group are successfully synthesized through spray drying and annealing methods. Density functional theory (DFT) calculations reveal that Mo6+ doping facilitates the transition of electrons from the valence to the conduction band, thus enhancing the intrinsic electron conductivity of Mox-NFPP. With an optimal Mo6+ doping level of x = 0.10, the Mo0.10-NFPP exhibits lower charge transfer resistance, higher sodium ion diffusion coefficients, and superior rate performance. As a result, the Mo0.10-NFPP cathode offers an initial discharge capacity up to 123.9 mAh g−1 at 0.1 C, nearly reaching its theoretical capacity. Even at a high rate of 10 C, it delivers a high discharge capacity of 86.09 mAh g−1, maintaining 96.18% of its capacity after 500 cycles. This research presents a new and straightforward strategy for enhancing the electrochemical performance of NFPP cathode materials for sodium-ion batteries.”

2. The TOC does not reflect the novelty and main findings of this study please m

Response 2: Thank you for your suggestion. The TOC image and related description are revised and have added in the revised manuscript on pages 1 and 2. The description for TOC image is as follows:

“A pure-phase Mo6+ doped Na4Fe3-xMo0.1(PO4)2P2O7/C composite is successfully synthesized using spray drying and annealing methods, which not only boosts the inherent electron conductivity, but also minimizes charge transfer resistance and speeds up Na+ ion transport. As a result, the electrochemical performance of this cathode material in sodium-ion batteries is significantly improved.”

3. The resolution of all figures should be improved significantly

Response 3: Thank you for your suggestion. All the figures within the revised manuscript have been updated with higher resolution.

4. The authors should rewrite the introductions section and improve the literature review. Also there are many sentences without cited references. The objectives of this study should be emphasized at the end of the introduction

Response 4: Thank you for your valuable advice. In the manuscript, we have revised the introduction section to include a comprehensive overview of the research background, advancements in NFPP materials, ion-doping strategies, the benefits of Mo-doping, the concept of this work, and our standout results. All the revisions have been marked in yellow color in pages 2 and 3.

5. The authors should carefully revise the scale bars, magnifications and spelling in Figure 2. For example ‘’um’’ should be ‘’µm’’. Also, please revise the materials abbreviations in the caption

Response 5: Thank you for your suggestion. As suggested, we have updated the spelling, scale bars, magnifications, and abbreviations in Figure 2 along with its caption.

6. The authors should mention the source of materials and precursors  

Response 6: Thank you for your suggestion. As recommended, we've incorporated in the experiment section all the sources and parameters for the materials and precursors utilized in this work. All the revisions have been marked in yellow color in page 3.

7. In Figure 3a, at the x-axis, please replace ‘’o’’ with ‘’angle’’. also the discussion related to the XRD data should be improved significantly and effect of doping should be highlighted. The diffraction peaks are the same with only slight shift, please explain.  

Response 7: Thank you for your valuable advice. We have modified the unit of x-axis to 2 θ/degree in Figure 3a, 3b and discussed the XRD results with further analysis including Rietveld refinement (Figure S2) on page 5, lines 182-193. In Figure 3a, all the diffraction peaks of Mox-NFPP shift towards lower angels, compared to 0-NFPP. For instance, the (022) crystal plane’s characteristic peak in 0-NFPP is at 2θ = 32.2°, while in Mo1.5-NFPP it shifts to 2θ = 31.9° (Figure 3b). Bragg's Law suggests this shift towards smaller angles indicates an expansion of the lattice spacing. Given the smaller radius of Mo6+ ions (0.59 Å) compared to Fe2+ ions (0.78 Å), it is likely that the increase lattice spacing is due to non-equivalent substitution by Mo6+ doping. This results in the creation of vacancies for charge balance (J. Mater. Chem. A, 2018, 6(4), 1390-1396), and the increase of interplanar crystal spacing, which could improve Na+ ion transport. In addition, the XRD of Mox-NFPP was subjected to Rietveld refinement, with the fitting results displayed in Figure S2a-S2d and Table S1. The lattice parameters of b and volume (V) of the Pna21 phases for Mox-NFPP have both increased compared to the pristine 0-NFPP. These results further support the estimation that Mo ions enter the crystal lattice, leading to its expansion.

8. The authors should add the high-resolution XPS spectra of all elements and fit them to see the effect of doping on the valence states and show the element phases.

Response 8: Thank you for your suggestion. We have incorporated all the elements’ HR XPS spectra results in Figure S3 and Figure 3c, 3d. The discussion has also been supplemented on pages 5 and 6. The XPS survey of the Mo0.10-NFPP sample, as seen in Figure S3a, confirms the presence of Na, Fe, O, P, and Mo elements. This aligns well with the EDS mapping results showcased in Figure 2g. Figure S3b-S3d show that the Na 1s, O 1s and P 2p spectra are consistent with the XPS fitting results of NFPP material as detailed in Chemsuschem, 2021, 14, 5424-5433. As reported in J. Power Sources, 2022, 521, 230922-230933, the Fe2+ orbital splitting peaks locates at 725.4 and 711.4 eV in pristine NFPP material. Here, in the Mo0.10-NFPP material, the Fe 2p peaks shifts marginally to the lower binding energy levels of 724.71 eV and 711.23 eV (Figure 3c). This suggests that the divalent state of iron in Mo0.10-NFPP remains constant after Mo6+ doping. The Fe-O bonds are slightly weakened, due to the reason that Mo atoms have a stronger ability to confer charge to nearby O atoms than Fe atoms, which has been verified in the theoretical calculation in Figure 6. In Figure 3d, the main peaks corresponding to Mo 3d5/2 and Mo 3d3/2 are observed at 232.41 eV and 235.05 eV, respectively, confirming that the Mo element is in the +6 oxidation state (J. Mater. Chem. A, 2018, 6(4), 1390-1396), which means that Mo6+ ions do not participate in the electrochemical reactions during charging process.

9. A comparison table for the performance with previously published articles should be added.

Response 9: Thank you for your valuable advice. Table S3 lists the electrochemical performance of Mo0.10-NFPP material in this work compared to other ion-doped NFPP materials reported in the literatures [S1-S6]. We have also added a discussion on performance comparison in the revised manuscript on page 7, lines 241-250. For comparison, the selected materials are all Fe-site doped. All doped materials exhibit excellent cycling stability, which is attributed to the robust structural stability of polyanionic cathode materials. In terms of rate performance, the main influencing factors include not only the intrinsic conductivity of the NFPP material but also the electronic conductivity of the carbon material in the composite material. This is evident from the good rate performance of composite materials containing high conductivity rGO [S1] or CNT [S4]. However, rGO and CNT materials are high cost additives. Among the composite carbon materials made from glucose [S2], sucrose [S5], and citric acid [S6] pyrolysis, Mo0.10-NFPP demonstrates better performance than Mn2+ [S2], Mg2+ [S3], Cr3+ [S4] and Ti4+ [S5] doped materials. Although V3+ doping also improves remarkably the rate performance of NFPP material, its high toxicity limits its application [S6]. Therefore, for NFPP material, Mo6+ doping appears to be an effective modification method with promising application prospects and feasibility.

Table S3 Comparison of electrochemical performance of ions doped NFPP materials from different research.

Doped
ions

Conductive coating materials

Discharge

capacity (mAh g−1)

Capacity retention

Ref.

Mo6+

C

123.9 (0.1C)

95.71 (5C)

86.09 (10C)

96.18% (500 cycles, 10C)

This work

Mn2+

rGO

131.5 (0.1C)

97.3 (5C)

90.8 (10C)

91.6% (200 cycles, 2C)

96.7% (700 cycles, 5C)

97.2 (2000 cycles, 10C)

[S1]

Mn2+

C

119.6 (0.1C)

88.97 (5C)

81.66 (10C)

94.6% (1000 cycles, 10C)

[S2]

Mg2+

C

104 (0.05 A g-1)

~75 (5 A g-1)

90.4% (5000 cycles, 5 A g-1)

[S3]

Cr3+

C

CNT

105.44 (0.1C)

89 (5C)

120.64 (0.1C)

87.11 (20C)

91.97% (500 cycles, 10C)

92.37% (2000 cycles, 10C)

[S4]

Ti4+

C

105.3 (0.2C)

84.9 (5C)

80 (10C)

91.4% (200 cycles, 1C)

93.8% (2000 cycles, 10C)

[S5]

V3+

C

123.4 (0.1C)

105.7 (5C)

102.3 (10C)

81.65% (10000 cycles, 20C)

[S6]

References:

[S1] Li, X.Q.; Zhang, Y.; Zhang, B.L.; Qin, K.; Liu, H.M.; Ma, Z.F. Mn-doped Na4Fe3(PO4)2(P2O7) facilitating Na+ migration at low temperature as a high performance cathode material of sodium ion batteries. J. Power Sources 2022, 521, 230922-230933.

[S2] Tao, Q.D.; Ding, H.Y.; Tang, X.; Zhang, K.B.; Teng, J.H.; Zhao, H.M.; Li, J. Mn-doped Na4Fe3(PO4)2P2O7 as a low-cost and high-performance cathode material for sodium-ion batteries. Energy Fuels 2023, 37, 6230−6239.

[S3] Xiong, F.Y.; Li, J.T.; Zuo, C.L.; Zhang, X.L.; Tan, S.S.; Jiang, Y.L.; An, Q.Y.; Chu, P.K.; Mai, L.Q. Mg-doped Na4Fe3(PO4)2(P2O7)/C composite with enhanced intercalation pseudocapacitance for ultra-stable and high-rate sodium-ion storage. Adv. Funct. Mater. 2023, 33, 2211257-2211265.

[S4] Wang, G.H.; Chen T.T.; Chen J.; Zhang Z.Y.; Zhen C.; Han X.Y.; Li J.G. Preparation and electrochemical properties of Na4Fe3-xCrx(PO4)2P2O7/C@CNT cathode materials for sodium ion batteries. Mod. Chem. Ind. 2024, 44(5), 149-154.

[S5] Huang, L.; Liu, C.J.; Bao, L.; Chen, Y.; Jiang, Y.H.; Fu, X.C. Large scalable preparation of Ti-doped Na4Fe3(PO4)2P2O7 as cathode material for high rate and long-life sodium-ion batteries. ACS Appl. Energ. Mater. 2023, 6, 11541-11549.

[S6] Zhang, H.; Cao, Y.J.; Liu, Z.L.; Cheng, X.S.; Li, X.L.; Xu, J.; Wang, N.; Yang, H.; Liu, Y.; Zhang, J.X. Structurally modulated Na4−xFe3−xVx(PO4)2P2O7 by vanadium doping for long-life sodium-ion batteries. ACS Sustainable Chem. Eng. 2024, 12, 5310−5318.

10. The explanation and discussions related to EIS data should be revised carefully. For instance ‘’ Figure 4c, the charge transfer resistances (Rct) for’’ It is Nyquist plot. Also the authors should fit the EIS data and add the solution resistance (Rs), charge transfer resistance (Rct), Warburg impedance (Wd), and constant phase element (CPE).

Response 10: Thank you for your suggestion. The EIS results in Figure 4c have been fitted and the results of Rs, Rct, Wd, CPE are listed in Table S2. The analysis of the EIS data has been modified and improved in pages 6 and 7 as following:

“Electrochemical Impedance Spectroscopy (EIS) tests were performed on the synthesized Mox-NFPP materials, as illustrated in Figure 4c. The equivalent circuit model was inserted in Figure 4c, where Rs represents the ohmic resistance, and Rct reflects the charge transfer resistance. The semicircle in the high-frequency region of the EIS curves corresponds to the Rct element, whereas the slope in the low-frequency region is dictated by the Warburg diffusion of Na+ ions from the surface to the middle of the cathode particles. The fitting results outlined in Table S2 highlight that Rs decreases slightly when increasing the Mo6+ doping level in Mox-NFPP samples. However, Rct sees a significant drop. This decrease in Rct is linked to the non-equivalent substitution of Mo6+, causing vacancies that speed up the transport of ions and electrons, thus enhancing charge transfer reactions [21,24,26]. The decrease in Rct contributes to the improved capacity and rate performance of the Mox-NFPP samples. However, it's worth noting that the Rct for Mo0.15-NFPP is higher than for Mo0.10-NFPP. This could be due to the inactivity of Mo6+ ions during the charge-discharge processes, potentially causing performance to drop with excessive doping. Further cycling performance analysis (Figure 4d) demonstrates that the Mo0.10-NFPP achieves a capacity of 86.09 mAh g−1 at a 10 C rate. Even after 500 cycles, it retains 96.18% of its capacity, demonstrating its excellent structural stability.

Table S2 Impedance parameters of Mox-NFPP materials

Sample

0-NFPP

Mo0.05-NFPP

Mo0.10-NFPP

Mo0.15-NFPP

Rs (Ω)

6.50

5.60

4.50

4.00

Rct (Ω)

1485

1253

725

1144

W

(Ω s-1/2)

182

261

310

312

CPE

300 nF

α=0.82

300 nF

α=0.81

300 nF

α=0.80

300 nF

α=0.82

11. The literature review is not enough, and more references should be cited, like ‘’ Nanomaterials202212(16), 2825 & Energy Conversion and Management: X, 2024. 22: p. 100570

Response 11: Thank you for your suggestion. As suggested, we have updated the literature review to comprehensively discuss the benefits and potential of ions doping strategies for enhancing the electrochemical performance of NFPP composites. We also have detailed the Mo doping strategies used in a variety of lithium/sodium ion battery cathode materials to clarify the modification mechanism. Meanwhile, additional relevant literature, such as Nanomaterials, 2022, 12(16), 2825 [Ref. 1] & Energy Conversion and Management: X, 2024. 22: 100570 [Ref. 2] are included.

Round 2

Reviewer 3 Report

Comments and Suggestions for Authors

The article could be accepted